# The Plasma Spectroscopic Study of Dergaon Meteorite, India

**DOI:** 10.3390/molecules25040984

**Published:** 2020-02-22

**Authors:** Abhishek K. Rai, Jayanta K. Pati, Christian G. Parigger, Sonali Dubey, Awadhesh K. Rai, Balen Bhagabaty, Amulya C. Mazumdar, Kalpana Duorah

**Affiliations:** 1Department of Earth and Planetary Sciences, Nehru Science Centre, University of Allahabad, Allahabad 211002, India; abhishekraigeology@gmail.com (A.K.R.); jkpati@gmail.com (J.K.P.); 2National Center of Experimental Mineralogy and Petrology, 14, Chatham Lines, University of Allahabad, Allahabad 211002, India; 3Department of Physics and Astronomy, University of Tennessee/University of Tennessee Space Institute,411 B. H. Goethert Parkway, Tullahoma, TN 37388, USA; 4Department of Physics, University of Allahabad, Allahabad 211002, India; sonalidubey193.sd@gmail.com (S.D.); awadheshkrai@gmail.com (A.K.R.); 5Department of Geological Sciences, Gauhati University, Guwahati 781014, India; b_bhagabaty@gauhati.ac.in (B.B.); acmazumdar@gauhati.ac.in (A.C.M.); 6Department of Physics, Gauhati University, Guwahati 781014, India or

**Keywords:** Dergaon meteorite, calibration-free laser-induced breakdown spectroscopy, atomic spectroscopy, molecular spectroscopy, planetary geochemistry

## Abstract

Meteorites are the recoverable portions of asteroids that reach the surface of the Earth. Meteorites are rare extraterrestrial objects studied extensively to improve our understanding of planetary evolution. In this work, we used calibration-free laser-induced breakdown spectroscopy (CF-LIBS) to evaluate the quantitative elemental and molecular analyses of the Dergaon meteorite, a H 4-5 chondrite fall sample from Assam, India. Spectral signatures of H, N, O, Na, Mg, Al, Si, P, K, Ca, Ti, Cr, Mn, Fe, Co, Ni, andIrweredetected. Along with the atomic emission, this work reports the molecular emission from FeO molecules. The concentration of the measured elements obtained using CF-LIBS is in close agreement with earlier reports. The elements H, N, and O and their concentrations are estimated by using CF-LIBS for the first time. This study applies laser spectroscopy to establish the presence of Ni, Cr, Co, and Ir in meteorites. The elemental analysis forms the basis for the establishment of the potential molecular composition of the Dergaon meteorite. Moreover, the elemental analysis approach bodes well for in-situ analyses of extraterrestrial objects including applications in planetary rover missions.

## 1. Introduction

Meteorites are the recoverable portions of asteroids that reach the surface of the Earth. The meteorites that are observed to fall and are collected are classified as “falls”. The remaining extraterrestrial objects of unknown age are grouped under “finds”. Only less than 2% of all meteorites (iron, stony, and stony-irons) collected to date are classified as “falls.” The Dergaon meteorite is one such “fall”, observed on March 2, 2001, at 16:40 hrs (Meteoritical Bulletin, 2001) near Dergaon (96°46’48”; 26°46’32”) village in Assam State, India. It is an ordinary chondrite (H 4-5) representing shock stage S5 [1] and comprises about ten different types of chondrules [2,3] with a variable chondrule: matrix ratio (60:40–80:20) [3]. The minerals that are present include:olivine (Ol: ~Fo_80_-Fa_20_), orthopyroxene (Opx: ~En_82_-Fs_18_), clinopyroxene (Cpx: ~Wo_47_-En_46_-Fs_7_), plagioclase (Plag: Ab_87_-An_13_), chlorapatite, chromite, merrillite, kamacite-taenite, and troilite [3]. In addition to about 92%of the volume of mineral phases, glass constitutes about 8% of the total volume [3]. It is a relatively well studied chondritic meteorite [3,4,5,6,7,8,9,10] possessing some interesting characteristics, such as potassium-depletion associated with vesiculated feldspars and the presence of aliphatic hydrocarbons [8,9] and nano-diamonds [10], observed using conventional analytical techniques including Gamma Ray, FTIR, and Laser Raman. The meteorite was successfully analyzed by laser-induced breakdown spectroscopy (LIBS) [11,12,13,14,15,16]. The calibration-free LIBS (CF-LIBS) approach was used to determine the elemental composition of the Dergaon meteorite. The CF-LIBS method is used for analyzing samples such as soils and rocks [17]. Furthermore, CF-LIBS can also analyze meteorite samples that are not easy to treat by traditional analytical techniques.

Using this approach, the traditional calibration curve is not required to quantify the elements present in the sample. The results obtained from CF-LIBS are compared with the results obtained from Instrumental Neutron Activation Analysis (INAA), Atomic Absorption Spectroscopy (AAS) and Inductively Coupled Plasma–Atomic Emission Spectroscopy (ICP-AES) data [2], and X-ray fluorescence (XRF) data [9] from past literature. Thus, our experiment demonstrates that CF-LIBS is a rapid technique for quantitative analysis, showing the presence of heavy and light elements rapidly while conserving the valuable extraterrestrial sample for posterity. The siderophile element ratio, such as Ni/Cr, with or without the presence of the platinum group of elements (PGE), e.g., Ir, has been used to decipher the presence of extraterrestrial components(such as chondrite meteorites) [18,19,20,21,22]. An attempt is also made in this study to establish the extraterrestrial affinity of the Dergaon sample by using high nickel/chromium content and the presence of iridium (Ir) [21], using LIBS spectral signatures [22]. We have quantified the presence of characteristic elements in the above meteorite using the CF-LIBS technique.

## 2. Results and Discussion

The spectra of Dergaon chondrite were recorded in the range of 200 to 900 nm. Table 1 lists spectral lines for various elements that were identified using the NIST spectral database [23].

The spectral lines of the elements present in the LIBS spectra of the Dergaon meteorite are shown in Figure 1a–d. The distinct advantage of LIBS lies in its unique ability to detect heavy as well as light elements, as indicated in Figure 1.

The spectral lines of characteristic trace elements observed in the LIBS spectra of the Dergaon chondritic meteorite include Cr, Co, Ni, and Ir, as shown in Figure 1b. It is essential to mention that Cr, Co, and Ni are also found in mafic-ultramafic terrestrial rocks. But iridium (Ir), a platinum group element, very rarely occurs in ultramafic rocks on Earth and is used to trace extraterrestrial signatures [21]. However, the extraterrestrial samples, like chondritic meteorites, show Ni/Cr > 1 and may contain substantial Ir. Furthermore, the wavelengths of spectral lines for characteristic elements found in the Dergaon meteorite included Ni I (338.05, 341.34, 342.37, 349.29 and 352.454 nm), Co I (340.51, 344.364, 347.39and 349.56 nm), and Ir I (351.364 nm). Identical spectral lines of these elements are also observed in meteorites, recently analyzed with LIBS [16]. An earlier study of the Dergaon meteorite reported the presence of noble gases and some light elements as well [2]. At a suitable gate delay, the signature of molecular emissions is also expected to occurwith LIBS Our experimental results reveal that the concentration of Fe and O in the present sample arean appreciable amounts, i.e., ≅ 28% and ≅ 34%, respectively. The high concentration of these elements may result in the formation of molecular bands in the recorded spectra [24,25]. Thus, we have tried to identify the molecular band of FeO in the LIBS spectra of theDergaon meteorite. Figure 2 displays signatures of molecular bands observed in the range of 558 to 564 nm.The molecular spectra belongs to the orange band system of the FeO molecule [26].

For verification, the concentration ratio of Ni to Cr (i.e., Ni/Cr>1) was further investigated using a quantitative analysis of elemental constituents present in the Dergaon meteorite [18,19,20,21,22]. In the CF-LIBS approach, the intensity of the spectral lines of all the elements present in the material is used and its application involves three assumptions:
(i)optically thin plasma,(ii)stoichiometric ablation, and(iii)thermal equilibrium.

The three assumptions (i) to (iii) are discussed in the subsequent paragraphs.

### 2.1. Optically Thin Plasma

In CF-LIBS, the measurement of the absolute intensity of spectral lines is required, i.e., the area under the spectral lines profile needs to be determined. The spectral lines used for the measurement of intensity should be free from self-absorption. Thisself-absorption of the spectral line causes distortion of the line profilewhichusually causes errors in the determined area. These errors may lead to the wrong values of electron density and temperature. Therefore, the laser-induced plasma should be optically thin to avoid self-absorption. The plasma is optically thin if the intensity ratio of two interference-free spectral lines of the element that havenearly the same upper energy level is equal to the productof transition probability, statistical weight, and the inverse of the wavelength of these spectral lines. The intensity ratios I/I′ for the spectral lines of different species that arepresent in the recorded spectra of the Dergaon meteorite were calculated. We found that these ratios are close to $\;A_{ki}g_{k}\lambda^{\prime }/A_{ki}^{\prime }g_{k}^{\prime }\lambda$. The results are summarized in Table 2. Consequently, the selected lines for analysis of the laser-induced plasma were optically thin [27,28].

### 2.2. Stoichiometric Ablation

The intensity of the spectral line present in the laser-induced plasma was directly related to the concentration of the element present in the sample. This was true when the composition of the laser-induced plasma was representative of the target material, i.e., ablation is stoichiometric. It is already experimentally established [28] that when laser irradiance at the focal spot of the sample surface is higher than 10^9^ Wcm^−2^, nearly every nanogram material from the focused spot explodes before the surface layer can vaporize. Therefore, the rapidly heated exploded material in the plasma has the same composition as the target, which leads to stoichiometric ablation [29]. In the present experiments, the laser energy was 15 mJ and the diameter of the focal spot was about ≈ 12 μm, and calculated irradiance at the focal spot was found to be ≈ 3.7 × 10^12^ Wcm^−2^,whichcorresponds to the condition of stoichiometric ablation and suggests that the laser-induced plasma is stoichiometric as laser irradiance is greater than 10^9^ Wcm^−2^.

### 2.3. Local Thermodynamic Equilibrium (LTE)

The intensity of a spectral line,$I_{\lambda }^{ki},$ corresponding to the transition from the upper level ‘k’ to the lower level ‘i’, can be calculated using
(1)$$I_{\lambda }^{ki}=F\;C_{s}A_{ki}\frac{g_{k}exp\left\{-\frac{E_{k}}{k_{B}T}\right\}}{Q},$$
Here, $A_{ki}$ is the transition probability in sec^−1^, g_k_ is the degeneracy factor (dimensionless), C_s_ is the concentration of the emitting atomic species, Q is the partition function of that species at plasma temperature, k_B_ is the Boltzmann constant, E_k_ is the energy of the upper level, λ is the wavelength of the spectral line, and F is an experimental parameter which takes into account the optical efficiency of the collection system. Equation (1) maybe written as
(2)$$ln\frac{I_{\lambda }^{ki}}{A_{ki}g_{k}}=-\frac{E_{k}}{k_{B}T}+ln\frac{C_{s}F}{Q},$$

For the determination of the plasma temperature, a graph of $ln\frac{I_{\lambda }^{ki}}{A_{ki}g_{k}}$ vs. $E_{k}$ was drawn, viz. a Boltzmann plot was constructed as illustrated in Figure 3.

According to the McWhirter criterion [30], plasma is inLTE if the value of electron density satisfies
N_e_ (cm^−3^) > 1.6 × 10^12^ × [T(K)]^1/2^ × [∆E (eV)]^3^,(3)
where, T is the plasma temperature, N_e_ is the electron density, and ∆E is the energy difference between upper and lower level transitions. In the present work, T is calculated by the Boltzmann plot given in Figure 3. The plasma temperature inferred from the Boltzmannplot is T = 12,064± 912 K. The average electron density of the laser-induced plasma is determined by measuring the full-width at half-maximum (FWHM), Δλ_1/2,_ of the Stark-broadened line of calcium 422.6 nm, illustrated in Figure 4, and using
Δλ_1/2_ = 2w[N_e_ (cm^−3^)/10^16^].(4)

Here, w is the electron impact parameter. From Ref. [31]: w = 7.18 × 10^−3^Å. The FWHM is obtained from Figure 4, and the value of the electron density is N_e_ ≈ 1.54 × 10^18^ cm^−3^. Therefore, the experimentally determined value of N_e_ (≈10^18^ cm^−3^) is larger than the limit set by the necessary McWhirter criterium thus, the laser-plasma is in LTE.

### 2.4. Species Concentrations

The concentrations, C_s_, are determined in the context of CF-LIBS by using Equation(2) and the slope of the Boltzmann plot in Figure 3. The following method is applied:(1)Find the area under of measured spectral lines of elements.(2)Infer temperature by constructing a Boltzmann plot.(3)Determine the intercept of the Boltzmann plot for each species.(4)Calculate the partition function for each species.(5)Evaluate the experimental efficiency factor, F, in Eq. (1), by normalizing the sum of the relative concentrations of all species in the sample, i.e., ∑Cs = 1(6)Finally, conclude species concentrations.

Table 3 communicates that the concentrations of Na, Mg, Al, Si, K, Ca, Ti, Cr, Mn, Fe, Co, and Ni determined in this work with CF-LIBS agree with the values reported using other methods, viz. INAA, AAS and ICP-AES, [2] and XRF [9] with deviations of the values within 10%. The total percentage of the elements determined by Shukla et al. and Saika et al. [2,9] is ≈ 47% and ≈ 66%, respectively. The reported element concentrations do not contain noble gases and light elements like H, N, and O. The LIBS spectra of the Dergaon meteorite recorded in ambient air show the presence of H, N, and O (see Figure 1d) unlike previous reports, although a minute atmospheric contribution cannot be ruled out. In the experiments, the interference of ambient air was suppressed by adjusting the distance between the lens and sample. The H, N, and O contents of the ambient air are 0.000055, 78.09, and 20.95%, respectively. However, the present data show that the concentration of O is 40 times larger than that of N(see Table 3), indicating a small spectral interference from the ambient air in the spectra. Importantly, this work is the first report showing the presence of light elements in the Dergaon meteorite. The concentrations of the Dergaon meteorite obtained by CF-LIBS are summarized in Table 3 and are in close agreement with the earlier data acquired by AAS, ICP-AES, XRF, and INAA [2,9].

## 3. Materials and Methods

The LIBS study was conducted on a sliced Dergaon fall meteorite sample, and five polished sections were used for petrographic study. Figure 5 illustrates the Dergaon sample. The LIBS spectra were obtained on a freshly cut even surface.

### Experimental Arrangement

The experimental setup is similar to our recent study [22]. The Nd: YAG laser radiation (Continuum Surellite III-10) was adjusted to an energy/pulse of 15 mJ at 532 nm. The pulse duration was 4 ns. A repetition rate of 10 Hz was selected for the recording of LIBS spectra from the meteorite. Every spectrum presented in this study corresponds to the average of 50 laser shots taken at 10 different places on the sample. The laser beam was directly focused onto the flat, sliced surface of the Dergaon meteorite sample (see Figure 1) using a convergent lens with a focal length of 15 cm to produce laser-induced plasmaat the surface of the sample. Care was taken to avoid crater formation on the sample surface. The meteorite sample was placed on atranslation stage and was continuously moved to get a fresh surface for each laser shot and to cover a suitable area. The emitted light from the laser-induced plasma was collected and fed into the Mechelle spectrometer (ME5000, Andor Technology, South Windsor, CT, US) equipped with an intensified charge-coupled device (ICCD) detector (model iStar 334, Andor Technology, South Windsor, CT, USA). The experimental parameters, especially the gate delay and gate width, were optimized to obtain the best S/B and S/N ratios of 0.7 μs and 4 μs, respectively.

## 4. Conclusions

The experimental investigation shows that laser-induced breakdown spectroscopy allows one to perform qualitative and quantitative elemental analysis of extraterrestrial materials with negligible sample loss. This is the first application of the calibration-free LIBS for a quantitative analysis of the Dergaon fall meteorite. The elemental concentration determination is similar to that obtained by conventional but time-consuming and expensive techniques, that cause significant sample destruction compared meager ablation loss observed during LIBS analysis. Identical to the previous study, the Dergaon meteorite shows depleted K concentration and Na concentrations much larger than K concentrations. It is an inherently K-depleted H+chondrite [2] rather than the previously suggested impact-induced high-temperature alkali devolatilization process [3], since the laser-induced breakdown spectroscopy applied in the present study did not alter the inferred concentrations of alkali elements.The elevated Mg (13.44%wt), Ni (1.30%wt), and Cr (0.55%wt) contents and confirmed presence of Ir (the spectral line at 351.364 nm) analogous to previous studies [2,9,16], and the petrographic observations establish the Dergaon sample as a chondrite meteorite. In addition to the characteristic atomic lines of the elements that were recorded, the spectra also show molecular bands of molecules composed of elements that display relatively higher concentration.

## Figures and Tables

**Figure 1 molecules-25-00984-f001:**
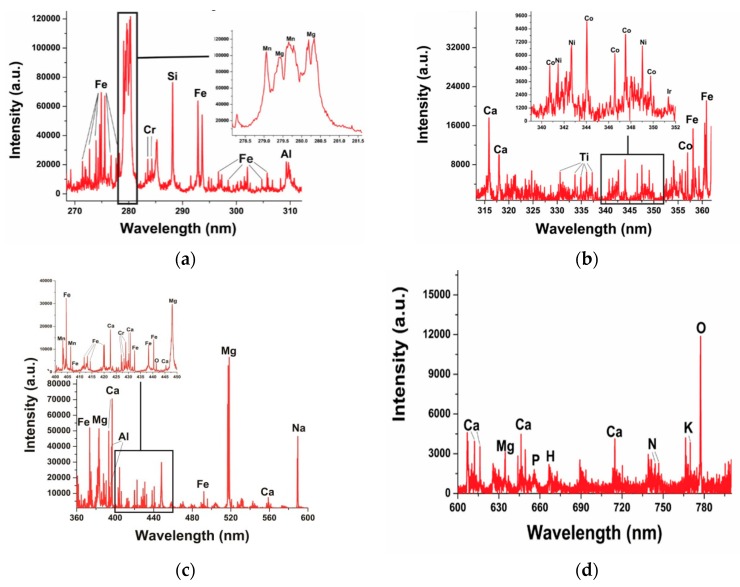
Typical laser-induced breakdown spectroscopy (LIBS) spectra of Dergaon meteorite in the wavelength range **(a)** 270–310 nm; **(b)** 315–360 nm; (**c**) 360–560 nm; **(d)** 600–800 nm.

**Figure 2 molecules-25-00984-f002:**
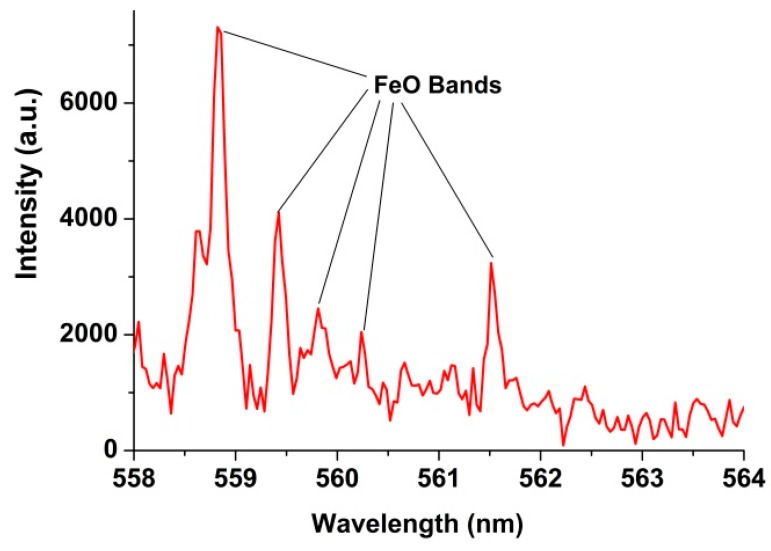
FeO molecular bands recorded with LIBS of the Dergaon meteorite.

**Figure 3 molecules-25-00984-f003:**
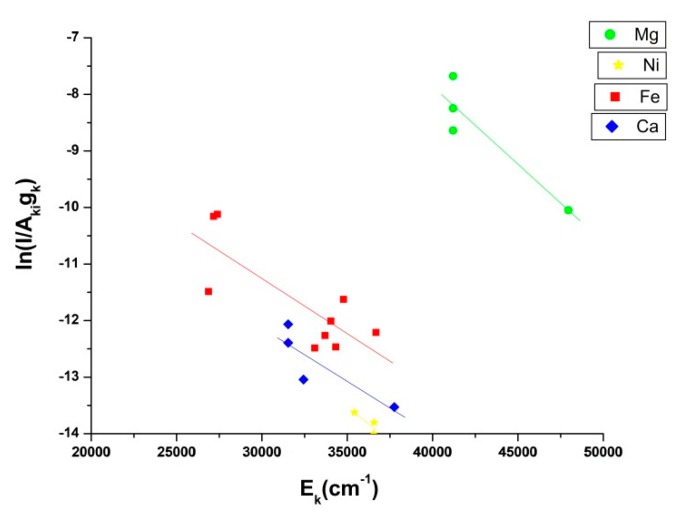
Boltzmann plot for different emission lines of elements for the Dergaon meteorite.

**Figure 4 molecules-25-00984-f004:**
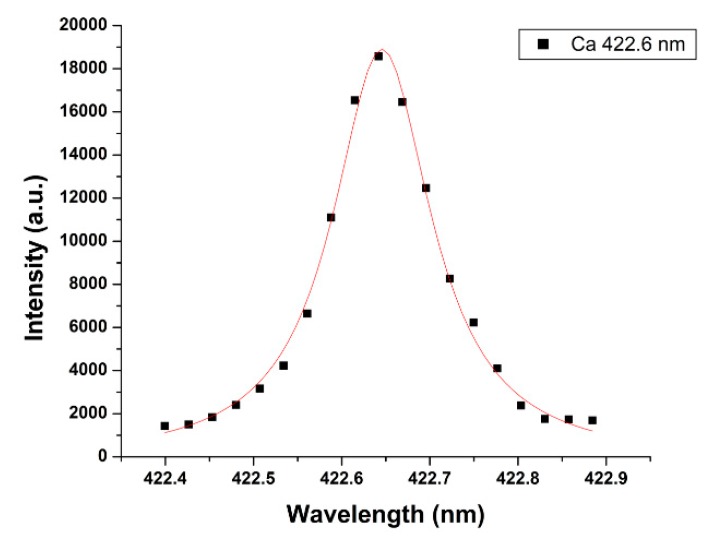
Measured and fitted Lorentzian of the Ca 422.6-nm line of the Dergaon meteorite.

**Figure 5 molecules-25-00984-f005:**
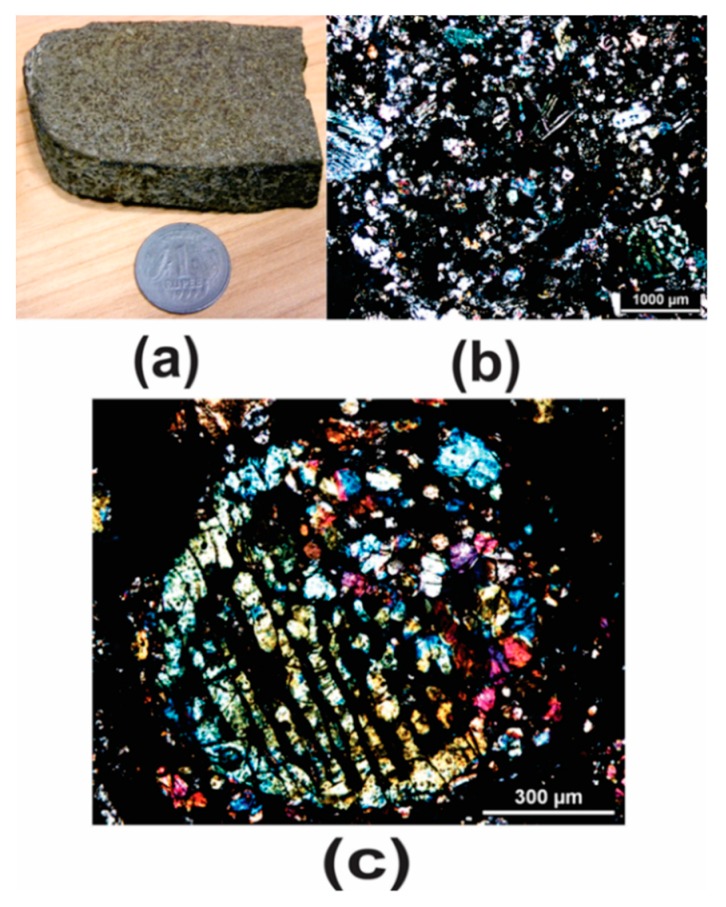
**(a)** Photograph of the Dergaon fall meteorite sample. **(b)** Photomicrograph of the Dergaon meteorite showing chondrules of various types and exhibiting porphyritic texture under crossed polarizers **(c)** A barred olivine chondrule from the Dergaon meteorite showing the olivine bars and a single rim in optical continuity. The glass phase is optically isotropic under crossed polarizers. Metallic phases are optically opaque.

**Table 1 molecules-25-00984-t001:** Spectral lines observed with laser spectroscopy of Dergaon meteorite.

Species	Major Lines (nm) Present in Dergaon Meteorite
H (1)	656.3(I)
N (7)	744.2(I), 746.8(I), 868.3(I)
O (8)	777.4(I), 844.6(I), 926.6(I)
Na (11)	588.9(I), 589.5(I)
Mg (12)	279.0(II), 279.5(II), 280.2(II),285.2(I), 382.9(I), 383.2(I), 383.8(I), 448.1(II), 516.7(I), 517.2(I), 518.3(I)
Al (13)	308.2(I), 309.2(I), 394.3(I), 396.1(I)
Si (14)	220.7(I), 221.0(I), 221.6(I), 250.6(I), 251.4(I), 251.6(I), 251.9(I), 252.8(I), 288.1(I), 390.5(I), 413.1(II),557.6(II), 634.7(II)
P (15)	645.9(II), 650.3(II), 650.7(II)
K (19)	766.4(I), 769.8(I)
Ca (20)	315.8(II), 317.9(II), 370.6(II), 373.6(II), 393.2(II), 396.7(II), 370.5(II), 422.6(I), 430.2(I), 442.5(I), 443.9(I), 445.4(I), 501.9(II), 518.8(I), 558.7(I), 610.2(I), 612.2(I), 616.2(I), 645.6(II), 646.2(I), 649.5(I), 647.2(I), 720.1(I)
Ti (22)	334.9(II), 336.1(II)
Cr (24)	283.5(II), 284.3(II), 284.9.(II), 302.0(I), 359.3(I), 360.5(I), 369.5(I), 381.5(I), 385.5(I), 428.9(I)
Mn (25)	257.6(II), 279.4(I), 279.8(I), 280.1(I), 380.6(I), 403.0(I), 403.3(I), 403.4(I)
Fe (26)	234.3(II), 238.2(II), 239.5(II), 240.4(II), 249.3(II), 250.7(II), 252.4(I), 254.3(II), 256.7(II),
	258.5(II), 259.8(II), 259.9(II), 260.7(II), 261.1(II), 273.9(II), 274.9(II), 275.5(II), 293.6(I),
	305.7(I), 305.9(I), 344.0(I), 358.0(I), 358.1(I), 364.7(I), 371.9(I), 373.4(I), 373.7(I),
	374.5(I), 374.9(I), 375.8(I), 382.0(I), 385.9(I), 404.1(I), 404.5(I), 406.3(I), 432.5(I), 438.3 (I)
Co (27)	340.5(I), 344.3(I), 347.3(I), 349.5 (I)
Ni (28)	338.0(I), 341.3(I), 342.3(I), 349.2(I),352.4(I)
Ir (77)	351.3(I)

**Table 2 molecules-25-00984-t002:** Physical parameters required for the calculation of intensity ratios of the two atomic lines for different elements in the recorded spectra of the Dergaon sample.

Species	$A_{ki}g_{k}\lambda^{\prime }\;/A_{ki\;}^{\prime }g_{k}^{\prime }\lambda \;(\mathrm{Theoretical})$	Intensity Ratio, I/I′ (Experimental)
Fe-I (374.9/375.8)	1.59	1.43 ± 0.02
Ca-II (315.8/317.9)	0.57	0.55 ± 0.01
Ca-11 (393.3/375.8)	2.06	1.99±0.04
Mg-II (279.5/280.2)	2.05	2.09 ± 0.05
Cr-I (359.3/360.5)	1.30	1.22±0.07

**Table 3 molecules-25-00984-t003:** Element concentrations in wt % in the Dergaon meteorite from different measurements.

Species	INAA, AAS and ICP-AES * (wt %)	XRF **(wt %)	CF-LIBS *** (wt %)
H	-	-	0.27
N	-	-	0.69
O	-	-	33.50
Na	0.7061	0.670	0.77
Mg	13.6	14.26	13.44
Al	1.09	1.20	1.70
Si	-	17.30	17.42
K	0.03	0.067	0.09
Ca	1.08	1.19	1.77
Ti	-	0.04	0.07
Cr	0.3705	0.03	0.55
Mn	0.2329	-	0.31
Fe	27.3	27.73	27.90
Co	0.08	-	0.07
Ni	1.82	1.75	1.30

* Reference INAA, AAS and ICP-AES data from Ref. [2]. ** Reference (XRF) data from Ref. [9]. *** Concentrations from this work.

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
