# Peer review of "The Plasma Spectroscopic Study of Dergaon Meteorite, India"

_molecules, 2020, doi:10.3390/molecules25040984_

Round 1

Reviewer 1 Report

Review

Title: Plasma Spectroscopic Study of Dergaon Meteorite, India

Topic: Authors are using the Laser Induced Breakdown Spectroscopy for the analytical measurement of Dergaon Meteorite from India.

Generally it is an interesting manuscript.

I have the following comments regarding this manuscript:

Line 180, Page 7

for Stark broadening specify individual variables. Define the electron impact parameter for calcium line 422.6 nm. Define the source, from where you get it.

For simplification I would suggest to use the H Balmer alpha line, see the reference:

M.A. Gigosos et al. / Spectrochimica Acta Part B 58 (2003) 1489–1504

line 208, page 8

For the calculation of the Boltzmann plots define the atomic parameters considered for determination of plasma temperature including, spectral atomic or ionic lines, partition function for all elements included in the Table 3.

In the Table 3 the " (wt %) " should be next to

INAA, AAS, ICP-AES (wt %)

XRF (wt %)

CF-LIBS (wt %)

In case that you have detected these elements by LIBS

"Spectral signatures of H, N, O, Na, Mg, Al, Si, P, K, Ca, Ti, Cr, Mn, Fe, Co, Ni, Ir"

you would need to include all of them in the Table 3, right?

As you are explaining in the line 191 relative concentrations of all elements present in the sample should be equal to one. In Figure 4, you have shown the Boltzmann plot only for three elements, Fe, Mg and Ca.

How you obtained the concentrations of individual elements?

Include Boltzmann plots for other elements in the Table 3.

Specify the plasma temperature you obtained.

Intersect(s) and slope(s) of Boltzmann plots are important for the CF LIBS, as well as the precise calculation of plasma temperature including errors.

In line 193, page 7 "Table 3 communicates that the concentration of Na, Mg, Al, Si,

K, Ca, Ti, Cr, Mn, Fe, Co, and Ni, determined in this work with CF-LIBS..."

in line 208, page 8

Table caption " Table 3. Intensity ratios of two atomic lines for in the recorded LIBS spectra. "

Are you sure? Please correct.

If you observed the Ir, what are the concentrations of Ir, in case that you using CF-LIBS?

in line 226, page 8 " The elevated Mg (13.9 wt %), Ni (1.90 wt %) and Cr (0.45 wt %)... "

in the Table 3 you claim 13,44 for Mg and Ni is not specified in the Table 3. Cr is in the Table 3 0.55 please write it consistently.

in line 123, page 5

"Figure 3. FeO molecular bands recorded with LIBS of Dergaon meteorite."

Specify in the table the Orange bands and the references

Suggestions: I suppose that these are the characteristic spectra from Calcium atomic lines:

Ca I at 558.876 nm, Rel.Int. 27

Ca I at 559.012 nm, Rel.Int. 24

Ca I at 559.447 nm, Rel.Int. 26

Ca I at 559.849 nm, Rel.Int. 25

Ca I at 560.129 nm, Rel.Int. 24

Ca I at 560.285 nm, Rel.Int. 24

line 108, Page 4

regarding Ir I (351.364 nm)

you should show this line,

I suppose due to the high content of Fe in your sample it is the iron atomic line:

Fe I at 351.38 nm Rel.Int. 191000

iron lines are quite strong and overlaps many components, especially with the low content and thus the emission from such peaks.

I would suggest, consult your results with the Prof. Parigger.

Author Response

Dear reviewer,

Please find our responses to your well-appreciated comments. Changes in the manuscript in response to reviewers are highlighted in green.

Respectfully,

CP

Reviewer 2 Report

The paper presented focuses on the study of rare extra-terrestrial objects, namely Dergaon meteorite. In the first part, the reader is provided with a brief introduction focusing mainly on classification and structural description of meteorites. Later the authors present an experimental methodology of calibration-free (CF) LIBS and required sample preparation. Then the achieved results are summarized in a table showing all detected elements together with their corresponding emission wavelengths. This is followed by a discussion, where all three conditions for performing CF-LIBS are introduced, verified and fulfilled. As the main outcome, the author presents a table summarizing all the results, the concentration of detected elements is given. The values of concentration are then confronted with the results acquired by the use of other techniques. The author then concludes by providing highlights and uniqueness of the applied method for this type of study. 

The novelty of this paper is that it is the first study focusing on the detection of light elements in the Dergaon meteorite. The strength of this work is mainly in its ability to present novel outcomes and confront them with the results already obtained by the use of other techniques.  

One of the weaknesses of the work presented could be the assumption of the sample homogeneity, leading to collection of spectra just from one plane of cut. The authors also do not provide how big was the area of the sample from which the spectra were collected.

Also, the concentration assumption is not really justified, expecting the sum of concentrations of selected elements is equal to one, omitting the concentration of remaining and not selected elements.

Overall the paper presents new results that can help shed light on some petrographic questions regarding the study of extraterrestrial objects. Also, the topic is quite interesting to a wide audience. In addition, the focus of the paper goes well with the scope of the selected journal.

Author Response

(The authors gave the same response as above.)

Reviewer 3 Report

I have read this manuscript with interest, since studies dealing with the analysis of meteorites are relatively rare in the LIBS literature. However, the content of this study was a let-down. I regret to say, but this study has minimum novelty or scientific value, but many problems. All findings and results of the study could be predicted or was known apriori. So my firm suggestion is reject. Please find my comments below.

The elemental composition of this particular meteorite has already been studied 10-15 years ago in details (see ref 2. or the data in Table 3); hence there is no new finding in terms of elemental composition at all. The considerable number of mineral components present in this meteorite have also been described in the literature in detail, and the authors themselves list these in the introduction. The „texture” or heterogeneity of the material can also be seen in Figure 1. Yet, the authors chose not pay any attention to this fact and simply collected 10-shot LIBS spectra on just any location on the sample surface. This nullifies the credibility or usefulness of any elemental concentration data determined, since at least the minor components are obviously present in different concentrations in different  minerals (that is why they are different minerals) – so should the analysis spot moved by a few mms, the results could be largely different! The lack of analytical experience is also caught by the fact that the other analytical techniques in Table 3 (e.g. AAS, ICP-AES, XRF, NAA) provide averaging, bulk concentrations, whereas LIBS gives localized (micro-spot) data. A reasonable attempt to obtain an average, bulk composition with LIBS should have involved thousands of laser shots and averaging - but then some of the potential added information of LIBS data would be lost        Sample characteristics also directly influence plasma properties, hence any temperature and Ne data, which are essential for CF-LIBS calculations, can also not be taken seriously, since the authors did not care about the heterogeneity of the sample. The experimental planning was also sub-adequte in other aspects as well. The experimental section reveals that a single collection lens was used to focus the beam to "a ca. 12 microns spot". This is impossible; such tight focusing can only be done with a compound lens system (e.g. microspot objective) nad/or with an aperture. All LIBS spectra were collected under room air, yet H, O and N concentrations are also reported… Although the sample is an extraterrestrial one, but the gate width and delay values were not optimized, just proclaimed to be 4 and 0.7 us. The authors make confusing and uninsighted arguments to justify the laser focusing settings they used. One prominent example is that they claim it to be important „not to generate an ablation crater” in order to conserve the sample, but at the same time they used a largely invasive sample preparation (cut and polished slab of the sample)… Generating an air plasma over the sample surface, as it was used here, just does not make any sense in LIBS, especially not in CF-LIBS. First, the emission spectrum will then be mostly representative of the air composition and not of the sample composition, not to mention T and Ne values. Second, if there is no ablation of the sample („no ablation crater”) then how could it be expected that LIBS analysis of the sample took place at all? Third, if there is no ablation, then how could it be stoichometric? (section 3.2.). The plasma diagnostics carried out on the (air) plasma is again a strange combination of meaningful and meaningless aspects. Section 3.1.: how could the plasma be optically thin overall, if there is 27.9% Fe, 17.4% Si, 13.4% Mg etc. in the samples? Sure, there could be some spectral lines for some elements found which may not be saturated, but overall, optical thickness can be expected. Hereby I would also like to point out to that the analytical lines selected for CF-LIBS (or details of the calculations) are not identified in the paper. Section 3.2.: the speculation about the stoichiometric ablation is week and does not justify anything… it is well known since long that ns LIBS tends to give biased results due to the different volatility of elements, plus the stoichiometric ablation condition should be experimentally checked for an extraterrestrial sample and for an „air plasma”. Section 3.3.: LTE in the LIB plasma has been studied hundreds of times in the literature and most studies usually come to the very convenient conclusion that there is LTE (otherwise no equilibrium plasma diagnistics or spectrum modeling could not be done), whereas some studies point out to the fact that LTE may only exist for a limited time window, depending on the experimental conditions…. I would also like to point out that LTE and the accurate knowledge of the plasma temperature are crucial for CF-LIBS. Yet, the authors do not disseminate any actual temperatures, but judged by what is in Figure 4., these must have been great inaccuracies that can not give only 10% error in concentrations in the end (as claimed in section 3.3.; besides, no standard deviations are provided in Table 3!), especially as the temperature is expected to depend on the mineral analyzed. Why are Ni, Ir and P LIBS results not reported in Table 3, although they were detected in the samples? (Table 1 and Figure 2), and Ni data is used in the discussion and conclusions?. Why is Ir only detected on one spectral line? In a complex sample, it is not a proof, and it is also an important element in the discussion… The organization of the Figures and Sections is also poor. It is unnecessary to place a „Dergaon meteorite” label on all graphs and tables, since it is the only sample studied. The headers of tables are confusing: if „Elements” is the header above a column, then the comun should only contain chemical symbols (not wavelengths or number of lines, as in Table 1 and 2), „Elements (wt%)” is also a poor label, when all concentration data are in other columns (Table 3). Figure 3 has no information content at all; those wide lines could be just coalesced atomic/ions lines… and this is the only "proof" that molecular bands were observed. A separate section should be devoted to the discussion of the concentration results – although not much of it is justified. The scientific expressions used are inaccurate at many places. Some examples: LIBS is just referred to as „laser spectroscopy” on many occasions (e,g, Table 1 caption, conclusions, introduction, etc.), spectral signatures are detected not „measured” (Abstract). The caption of Table 1 should be rephrased, as not „spectral wavelengths” were observed, but spectral lines. The last sentence of the experimental section makes no sense, it should be rephrased. etc.

Author Response

(The authors gave the same response as above.)

Reviewer 4 Report

Reviewer’s Comments on

Plasma Spectroscopic Study of Dergaon Meteorite, India

Abhishek K. Rai, Jayanta K. Pati, Christian G. Parigger, Sonali Dubey, Awadhesh K. Rai,

Bhagabaty, A.C. Mazumdar, and K. Duorah

Submitted to Molecules

This manuscript makes a significant contribution to the literature and should be published after minor revision. The revisions suggested below either involve minor grammatical changes or changes that will make the significance of the paper easier to be understood by readers.

Suggested changes:

Lines 2-3: Insert “the” into title so title is now “Plasma Spectroscopic Study of the Dergaon Meteorite, India” Lines 19-20: Change to “… asteroids (which normally occur between Mars and Jupiter within the solar system) that reach …” Line 21: Insert “the” so now reads “… to improve the understanding …” Line 24: Insert “and” and remove comma so now reads “… Ni, and Ir are measured.” Line 28: Insert “the” so now reads “… establish the presence …” Line 30: Change “in-situ” to “in situ” (which is italicized) Line 40: Insert comma and remove an existing comma so now reads ‘such “fall”, observed on March 2, 2001 at …’ Line 43: Insert space before “The” Line 49: Insert comma after “[8,9]” Line 50: Change to read “… Ray, FTIR, Laser Raman, etc.) Meteorites have been successfully …” Line 63: Remove “the” and insert “the” so now reads “… for posterity. … affinity of the Deragon …” Lines 63-65: The following sentence is not clear as to what it is trying to say. Include enough details so that it can be understood—the authors are addressing two audiences in this paper (the astrochemistry community and the laser spectroscopy community); as written the sentence is not intelligible at least for the laser spectroscopy community: “An attempt is also made in this study to establish the extraterrestrial affinity of the Deragon sample using high nickel/chromium content [18] and the presence of iridium (Ir) [18] based on LIBS spectral signatures [19].” What is “extraterrestrial affinity”? Are you tying to say that terrestrial and extraterrestrial samples can be differentiated based up the nickel-to-chromium ratio and on the presence of iridium? If so, please explicitly say so. Line 72: What does “KD” stand for? Line 73: Insert space between “types” and “and” Line 75: Change “polars” to “polarizers” Line 80: Change “amounted to” to “was” Line 80: Remove “of” so now reads “… recording LIBS spectra … Line 81: Change “average spectra” to “averaged spectrum” Line 82: Change “on” to “onto” Line 83: Change “plasma” to “plasmas” Line 85: Change “the” to “a” so now reads “… on a translation stage …” Line 86: Change “emiited” to “emitted” Line 86: Change “… fed to the Mechelle …” to “fed into a Mechelle …” Line 87: Insert “detector” so now reads “… intensified charge-coupled device (ICCD) detector …” Line 89: Insert “the” so now reads “… from the background …” Line 90: Insert hyphens so now reads “… signal-to-background and signal-to-noise …” Line 93: Insert “the” so now reads “… using the NIST …” Lines 95 and 101: Insert “the” so now reads “… spectra of the Dergaon …” Line 102: Insert comma after “Ir” Line 103: Remove comma after “But” Line 109: Insert “the” so now reads “… on the Dergaon …” Line 110: Insert comma after “gate delay” Line 111: Omit “the” and change “emission” to “emissions” so now reads “… of molecular emissions are …” Line 113: Insert comma before “respectively” Line 114: Change “(23)” to “[23]” Lines 115 and 123: Insert “the” so now reads “… spectra of the Dergaon …” Line 125: Change to “… ratio of Ni to Cr (i.e., Ni/Cr > 1) is further …” Line 137: Change “leads” to “lead” Line 138: Insert “the” so now reads “Therefore, the laser-induced …” Line 140: Change “multiplication” to “product” Line 141: Insert symbols to define transition probability, statistical weight, and wavelength. Since the authors are writing for both astrochemistry and laser spectroscopy communities, need to define these quantities here so that astrochemistry community can understand the equation in line 143. Line 143: Insert “the” so now reads “… spectra of the Dergaon …” Line 158: Insert spaces before and after “which” and insert comma before “which” Line 174: Insert “the” so now reads “… for the Dergaon …” Line 175: Insert symbol “Ne” after “electron density” Line 178: Insert symbol “Δλ1/2” after “FWHM” Line 178: Insert hyphen so now reads “Stark-broadened” Line 179: Insert “by” so now reads “… given by” Line 185: Insert space between “422.6” and “nm” Line 185: Insert “the” so now reads “… of the Dergaon …” Line 190: Insert space between “equation” and “2” Line 194: Remove comma after “Ni” Line 194: Change “… ICP-AES) (Shukla et al.) [2] and (XRF) (Saikia et al.) [9] …” to “… ICP-AES [2] and XRF [9]) …” Line 197: Change “does” to “do” Line 198: Insert comma and remove parentheses so now reads “… elements, like H, N, and O.” Line 198: Insert “the” so now reads “… of the Dergaon …” Line 199: Insert “the” so now reads “… show the presence …” Line 200: Insert “a” so now reads “… although a minute …” Line 208: The table caption is wrong. Currently states results in table involve intensity ratio of two atomic lines, but does not identify which lines are involved in the ratio. The table is of relative concentrations (in units of wt%). Note that the units for the last three columns are wt%, but the first column (the list of elements) has NO units so “(wt%)” must be moved away from the first column and either simply be given in the table caption OR be listed as such for the last three columns. Line 209: Change “… (P.N. Skukla et al. 2005) [2].” to “… [2].” Line 210: Change “… (B.J. Saikia et al. 2009) [9].” to “… [9].” Line 215: Change “… (Shukla et al. and Saikia et al.) [2,9].” to “… [2,9].” Line 219: Remove “the” so now reads “… of calibration-free LIBS ..” Lines 219-220: Insert “the” so now reads “… of the Dergaon …” Line 220: Change so now reads “… determinations are similar to those obtained …” Line 221: Insert comma and change “including” to “requiring” so now reads “… techniques, requiring significantly …” Line 224: Change “… (Shukla et al.) [2] …” to “… [2]…” Line 225: Change “… process (Ray et al.) [3] …” to “… process [3] …” Line 228: Change “… studies (Shukla et al., Saikia et al., Senesi et al.) [2,9, 16], and …” to “… studies [2,9,16], and …” Line 249: Insert space between “G.D.” and “Spectroscopic” Line 259: Insert space between “N.C.” and “Spectroscopic” Line 262: Insert space between “spectroscopy” and “Meteorit.” and italicize “Meteorit.” Line 270: Insert space between “R.C.” and “Textural” Line 280: Insert space between “R.T.” and “Geochemical”

Author Response

(The authors gave the same response as above.)

Round 2

Reviewer 3 Report

The manuscript has been slightly improved, but its fundamental problems remain.